# An advantage based policy transfer algorithm for reinforcement learning with metrics of transferability

## Abstract

Reinforcement learning (RL) can enable sequential decision-making in complex and high-dimensional environments if the acquisition of a new state-action pair is efficient, i.e., when interaction with the environment is inexpensive. However, there are a myriad of real-world applications in which a high number of interactions are infeasible. In these environments, transfer RL algorithms, which can be used for the transfer of knowledge from one or multiple source environments to a target environment, have been shown to increase learning speed and improve initial and asymptotic performance. However, most existing transfer RL algorithms are on-policy and sample inefficient, and often require heuristic choices in algorithm design. This paper proposes an off-policy Advantage-based Policy Transfer algorithm, APT-RL, for fixed domain environments. Its novelty is in using the popular notion of "advantage" as a regularizer, to weigh the knowledge that should be transferred from the source, relative to new knowledge learned in the target, removing the need for heuristic choices. Further, we propose a new transfer performance metric to evaluate the performance of our algorithm and unify existing transfer RL frameworks. Finally, we present a scalable, theoretically-backed task similarity measurement algorithm to illustrate the alignments between our proposed transferability metric and similarities between source and target environments. Numerical experiments on three continuous control benchmark tasks demonstrate that APT-RL outperforms existing transfer RL algorithms on most tasks, and is 10% to 75% more sample efficient than learning from scratch.

## 1 Introduction

A practical approach to implementing reinforcement learning (RL) in sequential decision-making problems is to utilize transfer learning. The transfer problem in reinforcement learning is the transfer of knowledge from one or multiple source environments to a target environment (Kaspar et al., 2020; Zhao et al., 2020; Bousmalis et al., 2018; Peng et al., 2018; Yu et al., 2017). The target environment is the intended environment, defined by the Markov decision process (MDP) $\mathcal{M}_\mathcal{T} = \langle \mathcal{X}, \mathcal{A}, \mathcal{R}_\mathcal{T}, \mathcal{P}_\mathcal{T} \rangle$, where $\mathcal{X}$ is the state-space, $\mathcal{A}$ is the action-space, $\mathcal{R}_\mathcal{T}$ is the target reward function and $\mathcal{P}_\mathcal{T}$ is the target transition dynamics. The source environment is a simulated or physical environment defined by the tuple $\mathcal{M}_\mathcal{S} = \langle \mathcal{X}, \mathcal{A}, \mathcal{R}_\mathcal{S}, \mathcal{P}_\mathcal{S} \rangle$ that, if learned, provides some sort of useful knowledge to be transferred to $\mathcal{M}_\mathcal{T}$. While in general $\mathcal{X}$ and $\mathcal{A}$ could be different between the source and target environments, we consider *fixed domain* environments here, where a domain is defined as $\langle \mathcal{X}, \mathcal{A} \rangle$ and is identical in the source and target.

For effective transfer of knowledge between fixed domain environments, it behooves the user to have $\mathcal{R}_\mathcal{S}$ and $\mathcal{P}_\mathcal{S}$ that are similar to $\mathcal{R}_\mathcal{T}$ and $\mathcal{P}_\mathcal{T}$, by either intuition, heuristics, or by some codified metric. Consider the single source transfer application to the four-room toy example for two different targets (Fig. 1). The objective is to learn a target policy, $\pi_\mathcal{T}^*$, in the corresponding target environments, $\mathcal{T}_1$ and $\mathcal{T}_2$ in Fig. 1b and 1c respectively, by utilizing knowledge from the source task, $\mathcal{S}$ in Fig. 1a, such that the agent applies the optimal or near-optimal sequential actions to reach the goal state. Intuitively, target task $\mathcal{T}_1$ is more similar to the source task $\mathcal{S}$ when compared to target task $\mathcal{T}_2$, and thus we expect that knowledge transferred from $\mathcal{S}$ to $\mathcal{T}_1$ will be comparatively more useful. We propose a formal way to quantify the effectiveness of transfer through a *transferability evaluation metric*. Our proposed metric, shown as $\tau_1$ for $\mathcal{T}_1$ in Fig. 1d as a function

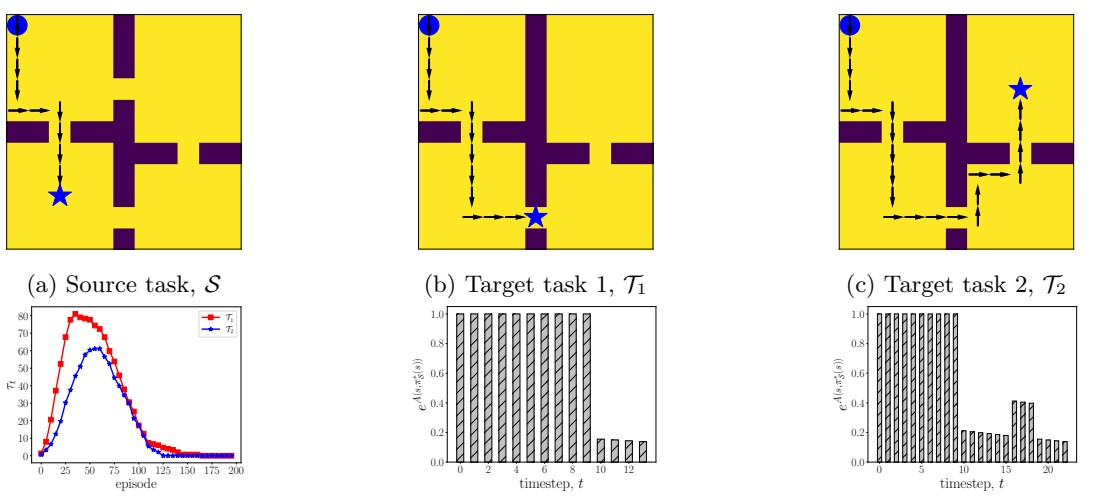

(a) Source task, $\mathcal{S}$      (b) Target task 1, $\mathcal{T}_1$      (c) Target task 2, $\mathcal{T}_2$

(d) Transferability evaluation metric    (e) Advantage based weight in $\mathcal{T}_1$    (f) Advantage based weight in $\mathcal{T}_2$

Figure 1: Knowledge transfer in the four-room toy problem: **Top row:** three tasks are presented in (a), (b) and (c), where ● represents the starting state and ★ represents the goal state; the goal state is moved further in (b) and (c) when compared to (a), and the doorways are also changed slightly. This makes the target task $\mathcal{T}_1$ in (b) more similar to the source task $\mathcal{S}$ in (a) when compared to the other target task $\mathcal{T}_2$ in (c). **Bottom row:** (d) An evaluation metric for transfer learning performance in tasks $\mathcal{T}_1$ and $\mathcal{T}_2$ is shown, which calculates performance at each evaluation episode, (e)-(f) influence of source policy on the target tasks $\mathcal{T}_1$ and $\mathcal{T}_2$ are shown in terms of $e^{A(s,a)}$ where $A(s,a) = Q^*_{\mathcal{T}_i}(s,a) - V^*(s)$ is the advantage function in task $\mathcal{T}_i$, and $i = 1, 2$. Note that the action is selected according to the source policy to calculate the advantage, which demonstrates the effect of the source policy on the target.

of evaluation episode $t$, indicates that the source knowledge from $\mathcal{S}$ is highly transferable to the target $\mathcal{T}_1$. In contrast, target task $\mathcal{T}_2$ is less similar to $\mathcal{S}$ and thus knowledge transferred from $\mathcal{S}$ to $\mathcal{T}_2$ may be useful, but should be less effective than that transferred to $\mathcal{T}_1$. Accordingly, the transferability evaluation metric $\tau_2$ for $\mathcal{T}_2$ in Fig. 1d indicates that the knowledge is useful ($\tau_2 > 0$), but not as useful as the transfer to $\mathcal{T}_1$ ($\tau_1 > \tau_2$). This task similarity measurement approach can provide critical insights about the usefulness of the source knowledge, and as we show, can also be leveraged in comparing the performance of different transfer RL algorithms. We also propose a theoretically-backed, model-based *task similarity measurement* algorithm, and use it to show that our proposed transferability metric closely aligns with the similarities between source and target tasks. Furthermore, we propose a new transfer RL algorithm that uses the popular notion of "advantage" as a regularizer, to weigh the knowledge that should be transferred from the source, relative to new knowledge learned in the target.

Our key idea is the following: calculate the advantage function based on actions selected using the source policy, and then use this advantage function as a regularization coefficient to weigh the influence of the source policy on the target. We can observe the effectiveness of this idea in our earlier toy example: Fig. 1e and 1f show the exponential of the advantage function as the regularization weight. By using the optimal source policy $\pi^*_{\mathcal{S}}$ to obtain an action, $a$, and calculating the corresponding advantage function $A(s,a)$, it is easy to see that $e^{A(s,a)}$ is lower in $\mathcal{T}_2$ when compared to $\mathcal{T}_1$. This result intuitively means that $\mathcal{S}$ can provide useful guidance for most of the actions selected by the target policy except the last 4 actions in $\mathcal{T}_1$, whereas, in contrast, the guidance is poor for the last 13 actions in $\mathcal{T}_2$. We show that this simple yet scalable framework can improve transfer learning capabilities in RL. Our proposed advantage-based policy transfer algorithm is straightforward to implement and, as we show empirically on several continuous control benchmark gym environments, can be at least as good as learning from scratch.

Our main contributions are the following:

- We propose a novel advantage-based policy transfer algorithm, APT-RL, that enables a data-efficient transfer learning strategy between fixed-domain tasks in RL (Algorithm 1). This algorithm incorporates two new ideas: advantage-based regularization of gradient updates, and synchronous updates of the source policy with target data.

- We propose a new *relative transfer performance* metric to evaluate and compare the performance of transfer RL algorithms. Our idea extends the previously proposed formal definition of transfer in RL by Lazaric (2012), and unifies previous approaches proposed by (Taylor et al., 2007; Zhu et al., 2020). We provide theoretical support for the effectiveness of this metric (Theorem 1) and demonstrate its use in the evaluation of APT-RL on different benchmarks and against other algorithms (Section 6).

- We propose a model-based *task similarity measurement* algorithm, and use it to illustrate the relationship between source and target task similarities and our proposed transferability metric (Section 6). We motivate this algorithm by providing new theoretical bounds on the difference in the optimal policies' action-value functions between the source and target environments in terms of gaps in their environment dynamics and rewards (Theorem 2).

- We conduct numerical experiments to evaluate the performance of APT-RL, and compare it against three benchmarks: zero-shot source policy transfer, SAC (Haarnoja et al., 2018) without any source knowledge, and REPAINT, a state-of-the-art transfer RL algorithm (Tao et al., 2020). We conduct these experiments on three dynamic systems which are created using several variations of the "inverted pendulum", "halfcheetah", and "ant" environments from OpenAI gym. We demonstrate that APT-RL outperforms existing transfer RL algorithms on most tasks, and is 10% to 75% more sample efficient than learning from scratch.

The paper is organized as follows: we begin by reviewing related work in section 2, We propose and explain our off-policy transfer RL algorithm, APT-RL, in section 3. Next, we propose evaluation metrics for APT-RL and a scalable task similarity measurement algorithm between fixed domain environments in section 4. We outline our experiment setup in section 5, and use the proposed transferability metric and task similarity measurement algorithm to evaluate the performance of APT-RL and compare it against other algorithms in section 6.

## 2 Related work

Traditionally, transfer in RL is described as the transfer of knowledge from one or multiple source tasks to a target task to help the agent learn in the target task. Transfer achieves one of the following: a) increases the learning speed in the target task, b) jumpstarts initial performance, c) improves asymptotic performance (Taylor & Stone, 2009). Transfer in RL has been studied somewhat extensively in the literature, as evidenced by the three surveys (Taylor & Stone, 2009; Lazaric, 2012; Zhu et al., 2020) on the topic.

The prior literature proposes several approaches for transferring knowledge in RL. One such approach is the transfer of instances or samples (Lazaric et al., 2008; Taylor et al., 2008). Another approach is learning some sort of representation from the source and then transferring it to the target task (Taylor & Stone, 2007b; 2005). Sometimes the transfer is also used in RL for better generalization between several environments instead of focusing on sample efficiency. For example, Barreto et al. (2017); Zhang et al. (2017) used successor feature representation to decouple the reward and dynamics model for better generalization across several tasks with similar dynamics but different reward functions. Another approach considered policy transfer where the KL divergence between point-wise local target trajectory deviation is minimized and an additional intrinsic reward is calculated at each timestep to adapt to a new similar task Joshi & Chowdhary (2021). In contrast to these works, our proposed approach simply uses the notion of advantage function to transfer *policy parameters* to take knowledge from the source policy to the target task and thus the transfer of knowledge is automated without depending on any heuristic choice.

There have also been different approaches to comparing source and target tasks and evaluating task similarity. A few studies have focused on identifying similarities between MDPs in terms of state similarities or policy similarities (Ferns et al., 2004; Castro, 2020; Agarwal et al., 2021). A couple of studies also focused on

transfer RL where each of the MDP elements is varying (Taylor & Stone, 2007a; Gupta et al., 2017). Most of these approaches either require a heuristic mapping or consider a high level of similarity between the source and target tasks. In contrast to these works, we develop a scalable task similarity measurement algorithm for fixed domain environments that does not require the learning of the optimal policy.

A few recent studies have focused on fixed domain transfer RL problems for high-dimensional control tasks (Zhang et al., 2018; Tao et al., 2020). Most of these studies are built upon on-policy algorithms which require online data collection and tend to be less data efficient than an off-policy algorithm (which we consider here). Although Zhang et al. (2018) discussed off-policy algorithms briefly along with decoupled reward and transition dynamics, a formal framework is absent. Additionally, learning decoupled dynamics and reward models accurately is highly non-trivial and requires a multitude of efforts. More recently, Tao et al. (2020) proposed an on-policy actor-critic framework that utilizes the source policy and off-policy instance transfer for learning a target policy. This idea is similar to our approach, but different in two main ways. First, we consider an entirely off-policy algorithm, unlike Tao et al. (2020), and second, our approach does not require a manually tuned hyperparameter for regularization. Additionally, Tao et al. (2020) discards samples collected from the target environment that do not follow a certain threshold value which hampers data efficiency. Finally, Tao et al. (2020) only considers environments where source and target only vary by rewards and not dynamics. In contrast, we account for varying dynamics, which we believe to be of practical importance for transfer RL applications.

## 3 APT-RL: An off-policy advantage based policy transfer algorithm

In this section, we present our proposed transfer RL algorithm. We explain two main ideas that are novel to this algorithm: advantage-based regularization, and synchronous updates of the source policy. As our proposed algorithm notably utilizes advantage estimates to control the impact of the policy from a source task, we call it Advantage based Policy Transfer in RL or **APT-RL** for short.

We build upon soft-actor-critic (SAC) (Haarnoja et al., 2018), a state-of-the-art off-policy RL algorithm for model-free learning. We use off-policy learning to re-use past experiences and make learning sample-efficient. In contrast, an on-policy algorithm would require collecting new samples for each gradient update, which often makes the number of samples required for learning considerably high. Our choice of off-policy learning therefore helps with the scalability of APT-RL.

### 3.1 Advantage-based policy regularization

The first new idea in our algorithm is to consider utilizing source knowledge during each gradient step of the policy update. Our intuition is that the current policy, $\pi_\phi$ parameterized by $\phi$, should be close to the source optimal policy, $\pi_{\mathcal{S}}^*$, when the source can provide useful knowledge. On the flip side, when the source knowledge does not aid learning in the target, then less weight should be put on the source knowledge. Based on this intuition, we modify the policy update formula of SAC as follows: we use an additional regularization loss with a temperature parameter, along with the original SAC policy update loss, to control the effect of the added regularization loss.

Formally, the policy parameter has the update formula:

$$\phi \leftarrow \phi + \alpha_\pi \left[ \hat{\nabla}_\phi J_1(\phi) + \beta \hat{\nabla}_\phi J_2(\phi) \right] \tag{1}$$

where $J_1(\cdot)$ is the usual SAC policy update loss, which uses soft-Q values instead of Q-values, and is parameterized by $\theta$ for the dataset $\mathcal{D}_{\mathcal{T}}$ and learning rate $\alpha_\pi$,

$$J_1(\phi) = \mathbb{E}_{\mathbf{s}_t \sim \mathcal{D}_{\mathcal{T}}} \left[ \mathbb{E}_{\mathbf{a}_t \sim \pi_\phi} [\alpha \log(\pi_\phi(\mathbf{a}_t|\mathbf{s}_t)) - Q_\theta(\mathbf{s}_t, \mathbf{a}_t)] \right], \tag{2}$$

and $J_2(\cdot)$ is the cross-entropy loss, $\mathcal{H}(\cdot, \cdot)$, between the source policy and the current policy,

$$J_2(\phi) = \mathcal{H}(\mu_\psi(\mathbf{a}|\mathbf{s}), \pi_\phi(\mathbf{a}|\mathbf{s})) = \mathbb{E}_{\mu_\psi(\mathbf{a}|\mathbf{s})}[-\log \pi_\phi(\mathbf{a}|\mathbf{s})], \tag{3}$$

where $\mu_\psi$ represents the optimal source policy $\pi_{\mathcal{S}}^*$ parameterized by $\psi$.

Thus, we are biasing the current policy $\pi_\phi$ to stay close to the source optimal policy $\pi_{\mathcal{S}}^*$ by minimizing the cross-entropy between these two policies while using the temperature parameter $\beta$ to control the effect of the source policy. Typically, this type of temperature parameter is considered a hyper-parameter and requires (manual) fine-tuning. Finding an appropriate value for this parameter is highly non-trivial and maybe even task-specific. Additionally, if the value of $\beta$ is not appropriately chosen, then the effect of the source policy may be detrimental to learning in the target task.

To overcome these limitations, we propose an *advantage-based* control of the temperature parameter, $\beta$. The core intuition of our idea is that the second term of Equation 1 should have more weight when an average action taken according to $\pi_\phi$ is better than a random action. If the source policy provides an action that is worse than a random action in the target, then $\pi_\phi$ should be regularized to have less weight. This is equivalent to taking the difference of the advantages based on the current policy and the source policy, respectively. In addition, we consider the exponential, rather than the absolute value, of this difference, so that the temperature approaches zero when the source provides adversarial knowledge.

Formally, our proposed advantage-based temperature parameter is determined as follows:

$$\beta_t = e^{A_{\mathcal{S}}^t - A_{\mathcal{T}}^t}, A_{\mathcal{T}}^t = Q_\theta(\mathbf{s}_t, \pi_\phi(\mathbf{s}_t)) - V(\mathbf{s}_t), A_{\mathcal{S}}^t = Q_\theta(\mathbf{s}_t, \mu_\psi(\mathbf{s}_t)) - V(\mathbf{s}_t) \tag{4}$$

Note that we can leverage the relationship between the soft-Q values and soft-value functions to represent the advantages from Equation 4 in a more convenient way that follows from (Haarnoja et al., 2018):

$$\begin{aligned} A_{\mathcal{T}}^t &= Q_\theta(\mathbf{s}_t, \pi_\phi(\mathbf{s}_t)) - \mathbb{E}[Q_\theta(\mathbf{s}_t, \mu_\psi(\mathbf{s}_t)) - \alpha \log \mu_\phi(\mathbf{a}_t|\mathbf{s}_t)] \\ A_{\mathcal{S}}^t &= Q_\theta(\mathbf{s}_t, \pi_\phi(\mathbf{s}_t)) - \mathbb{E}[Q_\theta(\mathbf{s}_t, \pi_\phi(\mathbf{s}_t)) - \alpha \log \pi_\phi(\mathbf{a}_t|\mathbf{s}_t)] \end{aligned} \tag{5}$$

### 3.2 Synchronous update of the source policy

We propose an additional improvement over the advantage-based policy transfer idea to further improve sample efficiency. As we consider a parameterized source optimal policy, $\mu_\psi$, it is possible to update the parameters of the source policy with the target data, $\mathcal{D}_{\mathcal{T}}$, by minimizing the SAC loss. The benefits of this approach is two-fold: 1) If the source optimal policy provides useful information to the target, then this will accelerate the policy optimization procedure by working as a regularization term in Equation 1, and 2) This approach enables sample transfer to the target policy. This is because the initial source policy is learned using the source data; when this source policy is further updated with the target data, it can be viewed as augmenting the source dataset with the latest target data.

Formally, the source policy will be updated as follows,

$$\psi \leftarrow \psi + \alpha_\psi \hat{\nabla}_\psi J_1(\psi) \tag{6}$$

where $J_1(\psi)$ is the typical SAC loss for the source policy with parameters $\psi$ and learning rate $\alpha_\psi$.

The pseudo-code for APT-RL is shown in Algorithm 1.

## 4 An evaluation framework for transfer RL

To formally quantify the performance of APT-RL, including against other algorithms, in this section, we propose notions of transferability and task similarity, as discussed in Section 1. First, we propose a formal notion of transferability, and use this notion to calculate a "relative transfer performance" metric. We demonstrate how this metric can be utilized to assess and compare the performance of APT-RL and similar algorithms. Then, we propose a scalable task similarity measurement algorithm for high-dimensional environments. We motivate this algorithm by providing a new theoretical bound on the difference in the optimal policies' action-value functions between the source and target environments in terms of gaps in their environment dynamics and rewards. This task similarity measurement algorithm can be used to identify

---

**Algorithm 1** APT-RL: **A**dvantage based **P**olicy **T**ransfer in **R**einforcement **L**earning

---

1: **Given:** parameterized source optimal policy, $\mu_\psi$, source learning step $\alpha_\mathcal{S}$
2: **Initialize:** current target policy, $\pi_\phi$, target buffer $\mathcal{D}_\mathcal{T} = \emptyset$
3: **for** each iteration **do**
4:     **for** each target environment step **do**
5:         $\mathbf{a} \sim \pi_\phi(\mathbf{a}_t|\mathbf{s}_t)$
6:         $\mathbf{s}' \sim p(\mathbf{s}'|\mathbf{s}, \mathbf{a})$
7:         $\mathcal{D}_\mathcal{T} \leftarrow \mathcal{D}_\mathcal{T} \cup \{(\mathbf{s}, \mathbf{a}, \mathcal{R}(\mathbf{s}, \mathbf{a}), \mathbf{s}')\}$
8:     **end for**
9:     **for** $G$ gradient updates **do**
10:         $\psi \leftarrow \psi + \alpha_\mathcal{S}\hat{\nabla}_\psi J_1(\psi)$ where $J_1$ is defined in Equation 2
11:         calculate $A_\mathcal{T}^t$ and $A_\mathcal{S}^t$ using Equation 5
12:         $\beta_t \leftarrow e^{A_\mathcal{S}^t - A_\mathcal{T}^t}$
13:         $\phi \leftarrow \phi + \alpha_\pi \left[\hat{\nabla}_\phi J_1(\phi) + \beta_t \hat{\nabla}_\phi J_2(\phi)\right]$
14:     **end for**
15: **end for**

---

| source knowledge, $\mathcal{K}_\mathcal{S}$ | target knowledge, $\mathcal{K}_{\mathcal{T},t}$ | evaluation metric, $\rho_t$ |
|---|---|---|
| samples, $\mathcal{D}_\mathcal{S}$ | samples, $\mathcal{D}_{\mathcal{T},t}$ | average returns, $G_t = \mathbb{E}^{\pi_{\mathcal{T},t}}\left[\sum_{k=0}^H r_k\right]$ |
| policy, $\mu_\psi$ | current policy, $\pi_{\mathcal{T},t}$ | number of samples to obtain reward threshold, $n_{\text{threshold}}$ |
| models $\mathcal{R}_\mathcal{S}, \mathcal{P}_\mathcal{S}$ | models, $\mathcal{R}_{\mathcal{T},t}, \mathcal{P}_{\mathcal{T},t}$ | area under the reward curve, $\Delta_t$ |
| value functions $Q_\mathcal{S}^*, V_\mathcal{S}^*$ | value functions $Q_{\mathcal{T},t}, V_{\mathcal{T},t}$ | number of samples to obtain asymptotic returns, $n_t$ |

Table 1: A list of potential source knowledge, target knowledge and evaluation metrics. This list unifies previous approaches proposed by (Taylor et al., 2007; Zhu et al., 2020). Note that target knowledge and evaluation metric is represented for the $t^{th}$ episode, and that $\pi_{\mathcal{T},t}$ denotes the optimal target policy at the evaluation episode $t$.

the best source task for transfer. Further, we use this algorithm in the experiment section, to illustrate how our proposed relative transfer performance metric closely aligns with the similarities between the source and target environment.

## 4.1 A metric of transferability

We formally define *transferability* as a mapping from stationary source knowledge and non-stationary target knowledge accumulated until timestep $t$, to a learning performance evaluation metric, $\rho_t$.

**Definition 4.1** (Single-task transferability). Let $\mathcal{K}_\mathcal{S}$ be the transferred knowledge from a source task $\mathcal{M}_\mathcal{S}$ to a target task $\mathcal{M}_\mathcal{T}$, and let $\mathcal{K}_{\mathcal{T},t}$ be the available knowledge in $\mathcal{M}_\mathcal{T}$ at timestep $t$. Let $\rho_t$ denote a metric that evaluates the learning performance in $\mathcal{M}_\mathcal{T}$ at timestep $t$. Then, transferability is defined as the mapping

$$\Lambda : \mathcal{K}_\mathcal{S} \times \mathcal{K}_{\mathcal{T},t} \to \rho_t \ .$$

Intuitively, this means that $\Lambda(\cdot)$ takes prior source knowledge and accumulated target knowledge to evaluate the learning performance in the target task.

As an example, if the collection of source data samples $\mathcal{D}_\mathcal{S}$ are utilized as the transferred knowledge and the average returns using the latest target policy, $G_t = \mathbb{E}^{\pi_{\mathcal{T},t}}\left[\sum_{k=0}^H r_k\right]$, is used as the evaluation performance of a certain transfer algorithm $i$, then the transferability of algorithm $i$ at episode $t$, can be written as the following, $\Lambda_i : \mathcal{D}_\mathcal{S} \times \mathcal{D}_{\mathcal{T},t} \to G_t$.

Notice that we leave the choice of input and output of this mapping as user-defined task-specific parameters. Any traditional transfer methods can be represented using the idea of transferability. Potential choices for source and target knowledge and evaluation metrics are listed in Table 1. Our idea extends the previously

proposed formal definition of transfer in RL by Lazaric (2012), and unifies previous approaches proposed by (Taylor et al., 2007; Zhu et al., 2020).

Expressing transfer learning algorithms in terms of this notion of transferability has a number of advantages. First, this problem formulation can be easily extended to unify other important RL settings. For example, this definition can be extended to offline RL (Levine et al., 2020) by considering $\mathcal{K}_\mathcal{S} = \mathcal{D}_{\text{source}}$, and $\mathcal{K}_{\mathcal{T},t} = \emptyset$, $\forall t$. Second, the comparison of two transfer methods becomes convenient if they have the same evaluation criteria. For instance, one way to construct evaluation criteria may be to use sample complexity in the target task to achieve a desired return. Subsequently, the transferability metric can be used to assess a "relative transfer performance" metric, which can act as a tool for comparing two different transfer methods conveniently.

**Definition 4.2** (Relative transfer performance, $\tau$). Given the transferability mapping of algorithm $i$, $\Lambda_i$, the relative transfer performance is defined as the difference between the corresponding learning performance metric $\rho_t^i$ and learning performance metric from a base RL algorithm $\rho_t^b$ at evaluation episode $t$. Formally,

$$\tau_t = \rho_t^i - \rho_t^b \ ,$$

where the base RL algorithm represents learning from scratch in the target task (meaning $\mathcal{K}_\mathcal{S} = \emptyset$), and $\rho_t^i$, $\rho_t^b$ are the same evaluation criteria of learning performances.

### 4.1.1 Theoretical support

We first formally show that, with an appropriate definition of the evaluation metric, non-negative relative transfer performance leads to a policy in the target task which is at least as good as learning from scratch.

**Theorem 1.** *(Relative transfer performance and policy improvement) Consider* $\rho_t^i = \mathbb{E}^{\pi_{i,t}}\left[\sum_{k=0}^H r_k|\mathbf{s}_0\right]$ *for policy* $\pi_i$ *and* $\rho_t^b = \mathbb{E}^{\pi_{b,t}}\left[\sum_{k=0}^H r_k|\mathbf{s}_0\right]$ *for policy* $\pi_b$*, where* $\mathbf{s}_0$ *is the starting state and each policy is executed for* $H$ *timesteps, then the learned policy,* $\pi_{i,t}$ *using algorithm* $i$*, in the target at episode* $t$ *is at least as good as the source optimal policy,* $\pi_{b,t}$ *if* $\tau_t \geq 0$*.*

The proof can be found in the Appendix **??**.

### 4.1.2 Revisiting the toy problem

We also leverage the toy example presented in Fig. 1 to explain the proposed concepts. The performance evaluation metric is chosen as the average returns, collected from an evaluation episode $t$, that is $\rho_t = \sum_{k=0}^H r_k$. For transfer, we choose the low-level direct knowledge Q-values. At first, we initialize the target Q-values with pre-trained source Q-values, $Q_\mathcal{S}^*$. Thus, at each episode, $t$, the updated Q-values in the target are a combination of both source and target knowledge. Thus, the transferability mapping can be expressed as $\Lambda_{\text{Q-learning}} : Q_\mathcal{S}^* \times Q_{\mathcal{T},t} \rightarrow \mathbb{E}^{\pi_{\mathcal{T},t}}\left[\sum_{k=0}^H r_k\right]$. We calculate $\rho_t$ after every 10 timestep by executing the greedy policy from the updated Q-values for a fixed time horizon. Relative transfer performance, $\tau_t$, remains non-negative for both the target tasks $\mathcal{T}_1$ and $\mathcal{T}_2$, for up to around 125 evaluation episodes. Intuitively this means that the learning performance of both policies is better than a base algorithm for all of the evaluation episodes. Also, $\tau_t$ is substantially higher for $\mathcal{T}_1$ than $\mathcal{T}_2$ which means that transferring knowledge from $\mathcal{S}$ leads to better learning performance in $\mathcal{T}_1$ compared to $\mathcal{T}_2$. This can be explained by the fact that the dynamics in $\mathcal{T}_1$ are more similar to $\mathcal{S}$ than $\mathcal{T}_2$.

## 4.2 Measuring task similarity

As seen in the toy problem above, a measure of task similarity would help us illustrate the close alignment of our proposed transferability metric with the similarities of the source and target tasks. Beyond this, measuring task similarity can provide additional insights into why a particular source task is more appropriate to transfer knowledge to the target task. Motivated by these, we propose an algorithm for measuring task similarity in this section.

### 4.2.1 Theoretical motivation

To motivate the idea behind our proposed algorithm, we first investigate theoretical bounds on the expected discrepancies between the policies learned in the source and target environments. One effective way for this analysis is to calculate the upper bound on differences between the optimal policies' action-values. Previously, action-value bounds have been proposed for similar problems by Csáji & Monostori (2008); Abdolshah et al. (2021). We extend these ideas to the transfer learning setting where we derive the bound between the target action-values under target optimal policy, $\pi_{\mathcal{T}}^*$ and target action-values under source optimal policy, $\pi_{\mathcal{S}}^*$.

**Theorem 2.** *(**Action-value bound between fixed-domain environments**) If $\pi_{\mathcal{S}}^*$ and $\pi_{\mathcal{T}}^*$ are the optimal policies in the MDPs $\mathcal{M}_{\mathcal{S}} = \langle \mathcal{X}, \mathcal{A}, \mathcal{R}_{\mathcal{S}}, \mathcal{P}_{\mathcal{S}} \rangle$ and $\mathcal{M}_{\mathcal{T}} = \langle \mathcal{X}, \mathcal{A}, \mathcal{R}_{\mathcal{T}}, \mathcal{P}_{\mathcal{T}} \rangle$ respectively, then the corresponding action-value functions can be upper bounded by*

$$||\mathbf{Q}_{\mathcal{T}}^{\pi_{\mathcal{T}}^*} - \mathbf{Q}_{\mathcal{T}}^{\pi_{\mathcal{S}}^*}||_\infty \leq \frac{2\delta_{\mathcal{ST}}^r}{1-\gamma} + \frac{2\gamma\delta_{\mathcal{ST}}^{TV}(R_{max,\mathcal{S}} + R_{max,\mathcal{T}})}{(1-\gamma)^2} \tag{7}$$

*where $\delta_{\mathcal{ST}}^r = ||\mathcal{R}_{\mathcal{S}}(\mathbf{s}, \mathbf{a}) - \mathcal{R}_{\mathcal{T}}(\mathbf{s}, \mathbf{a}))||_\infty$, $\delta_{\mathcal{ST}}^{TV}$ is the total variation distance between $\mathcal{P}_{\mathcal{S}}$ and $\mathcal{P}_{\mathcal{T}}$, $\gamma$ is the discount factor and $R_{max,\mathcal{S}} = ||\mathcal{R}_{\mathcal{S}}(\mathbf{s}, \mathbf{a})||_\infty$, $R_{max,\mathcal{T}} = ||\mathcal{R}_{\mathcal{T}}(\mathbf{s}, \mathbf{a})||_\infty$.*

The proof of this theorem can be found in the Appendix **??**. Note that as we propose APT-RL for fixed-domain environments, the bound can be expressed in terms of differences in the remaining environment parameters: the total variation distance between the source and target transition probabilities, and the maximum reward difference between the source and the target. Intuitively this bound means that a lower total variation distance between the transition dynamics can provide a tighter bound on the deviation between the action-values from the target and source optimal policies. Similarly, having a smaller reward difference also helps in getting lower action-value deviations in the target. Also note that, if the reward function or transition dynamics remain identical between source and target, then the corresponding term on the right side of Equation 7 vanishes.

Next, motivated by this bound, we propose a task similarity measurement algorithm that assesses the differences between source and target dynamics and rewards in order to evaluate their similarity.

### 4.2.2 A model-based task similarity measurement algorithm

Previous attempts for measuring task similarity include behavioral similarities in MDP in terms of state-similarity or bisimulation metric, and policy similarity (Ferns et al., 2004; Castro, 2020; Agarwal et al., 2021). Calculating such metrics in practice is often challenging due to scalability issues and computation limits. Additionally, our key motivation is to find a similarity measurement that does not require solving for the optimal policy *apriori*, as the latter is often the key challenge in RL. To this end, we propose a new model-based method for calculating similarities between tasks.

We propose an encoder-decoder based deep neural network model at the core of this idea. For any source or target task, a dataset, $\mathcal{D} = \{(\mathbf{s}, \mathbf{a}, r, \mathbf{s}')\}$, is collected by executing a random policy. Next, a dynamics model, $f_{\mathcal{P}}(\mathbf{s}, \mathbf{a})$, is trained by minimizing the mean-squared-error (MSE) loss using stochastic gradient descent.

$$\mathcal{L}_{\text{dyn}} = ||\mathbf{s}' - f_{\mathcal{P}}(\mathbf{s}, \mathbf{a})||_2 \tag{8}$$

Similarly, a reward model, $f_{\mathcal{R}}(\mathbf{s}, \mathbf{a})$ is trained using the collected data to minimize the following MSE loss.

$$\mathcal{L}_{\text{rew}} = ||r - f_{\mathcal{R}}(\mathbf{s}, \mathbf{a})||_2 \tag{9}$$

The encoder portion of the neural network model encodes state and action inputs into a latent vector. Then, the decoder portion uses this latent vector for the prediction of the next state or reward. We consider decoupled models for this purpose, meaning that we learn separate models for reward and transition dynamics from the same dataset. This allows us to identify whether only reward or transition dynamics or both vary between tasks. Once these models are learned, the source model is used to predict the target data and calculate the $L_2$ distance between the predicted and actual data as the similarity error. If $\xi_k^{\mathcal{P}}$ and $\xi_k^{\mathcal{R}}$ are

the similarity errors in target dynamics and rewards, respectively, we can calculate the dynamics and reward similarity separately as follows,

$$\text{dynamics similarity: } \Xi_{\mathcal{S},\mathcal{T}}^{\mathcal{P}} = \frac{1}{|\mathcal{D}_{\mathcal{T}}|} \sum_{k=1}^{|\mathcal{D}_{\mathcal{T}}|} \xi_k^{\mathcal{P}}, \text{reward similarity: } \Xi_{\mathcal{S},\mathcal{T}}^{\mathcal{R}} = \frac{1}{|\mathcal{D}_{\mathcal{T}}|} \sum_{k=1}^{|\mathcal{D}_{\mathcal{T}}|} \xi_k^{\mathcal{R}} \tag{10}$$

Our approach is summarized in Algorithm 2. This approach can be viewed as a modern version of (Ammar et al., 2014), but instead of using Restricted Boltzmann Machines, we use deep neural network-based encoder-decoder architecture to learn the models, and we do it in a decoupled way. Although we only consider fixed domain environments in this study, having an encoder-decoder model allows for measuring task similarity between tasks that may not have the same domain. This can be done by encoding the state-action input into a fixed-dimensional latent vector for both tasks. Another potential application of our idea can be multi-task transfer problems, where we want to find the source task that is most similar to the target for knowledge transfer.

---

**Algorithm 2** Model-based task similarity measurement

.

1: Collect $m$ data samples from source $\mathcal{M}_{\mathcal{S}}$ using a random policy, $\mathcal{D}_{\mathcal{S}} = \{(\mathbf{s}_{\mathcal{S}}, \mathbf{a}_{\mathcal{S}}, r_{\mathcal{S}}, \mathbf{s}'_{\mathcal{S}})\}$
2: Collect $m$ data samples from target $\mathcal{M}_{\mathcal{T}}$ using a random policy, $\mathcal{D}_{\mathcal{T}} = \{(\mathbf{s}_{\mathcal{T}}, \mathbf{a}_{\mathcal{T}}, r_{\mathcal{T}}, \mathbf{s}'_{\mathcal{T}})\}$
3: Learn dynamics models, $f_{\mathcal{P}}^{\mathcal{S}}(\mathbf{s}, \mathbf{a}), f_{\mathcal{P}}^{\mathcal{T}}(\mathbf{s}, \mathbf{a})$ using $\mathcal{D}_{\mathcal{S}}$ and $\mathcal{D}_{\mathcal{T}}$ and minimizing Equation 8
4: Learn reward models, $f_{\mathcal{R}_{\mathcal{S}}}(\mathbf{s}, \mathbf{a}), f_{\mathcal{R}_{\mathcal{T}}}(\mathbf{s}, \mathbf{a})$ using $\mathcal{D}_{\mathcal{S}}$ and $\mathcal{D}_{\mathcal{T}}$ and minimizing Equation 9
5: **for** each $(\mathbf{s}_{\mathcal{T}}, \mathbf{a}_{\mathcal{T}}, r_{\mathcal{T}}, \mathbf{s}'_{\mathcal{T}}) \in \mathcal{D}_{\mathcal{T}}$ **do**
6: $\quad \hat{\mathbf{s}}'_{\mathcal{T}} = f_{\mathcal{P}_{\mathcal{T}}}(\mathbf{s}_{\mathcal{T}}, \mathbf{a}_{\mathcal{T}}), \hat{r}_{\mathcal{T}} = f_{\mathcal{R}_{\mathcal{T}}}(\mathbf{s}_{\mathcal{T}}, \mathbf{a}_{\mathcal{T}})$
7: $\quad \xi_k^{\mathcal{P}} = ||\hat{\mathbf{s}}'_{\mathcal{T}} - \mathbf{s}'_{\mathcal{T}}||_2, \xi_k^{\mathcal{R}} = ||\hat{r}_{\mathcal{T}} - r_{\mathcal{T}}||_2$
8: **end for**
9: dynamics similarity, $\Xi_{\mathcal{S},\mathcal{T}}^{\mathcal{P}} = \frac{1}{|\mathcal{D}^{\mathcal{T}}|} \sum_{k=1}^{|\mathcal{D}^{\mathcal{T}}|} \xi_k^{\mathcal{P}}$
10: reward similarity, $\Xi_{\mathcal{S},\mathcal{T}}^{\mathcal{R}} = \frac{1}{|\mathcal{D}^{\mathcal{T}}|} \sum_{k=1}^{|\mathcal{D}^{\mathcal{T}}|} \xi_k^{\mathcal{R}}$

---

## 5 Experiment Setup

We apply the described methods to three popular continuous control benchmark gym environments (Brockman et al., 2016) to demonstrate performance on increasingly complex problems: 1) 'Pendulum-v1', 2) 'HalfCheetah-v3' and 3) 'Ant-v3' environments. The vanilla Gym environments are not effective for transfer learning settings. Thus we create four different perturbations of the vanilla Gym environments, with perturbations to dynamics made in all three environments and an additional reward perturbation in the Half Cheetah environment. Five different tasks are created for each experiment. The standard gym environment is considered the source task and the modified environments are considered the target tasks. We provide a high-level overview of each environment below. Details of the environments and the algorithm hyperparameters can be found in Appendices **??** and **??**.

**Pendulum-v1:** This is a continuous control problem to keep a pendulum in the inverted position by applying the appropriate torque. Each observation is a 3-dimensional vector of the x-y coordinates of the pendulum's free end and its angular velocity, and the action is a torque value between $[-2.0, 2.0]$. The source environment has a gravitational magnitude of $g = 10.0$ and the four target environments have $g = [12.0, 14.0, 16.0, 18.0]$. In this way, the most similar target task has a gravity of 12.0 and the least similar target has a gravity of 18.0.

**HalfCheetah-v3:** This is a complex continuous control task of a 2D cat-like robot where the objective is to apply a torque on the joints to make it run as fast as possible. The observation space is 17-dimensional and the action space, $a \in \mathbb{R}^6 \forall a \in \mathcal{A}$. Each action is a torque applied to one of the front or back rotors of the robot and can take a value between $[-1.0, 1.0]$. We make two types of perturbations to create target tasks; reward variation and dynamics variation. For the reward variation, the source environment uses a forward reward of $+1$, and the four target environments have a forward reward $r = [-2, -1, 1, 2]$. Note

that a negative forward reward is a target task where the robot needs to run in the opposite direction than the source. For this type of example, the source acts as an adversarial task and the goal is to learn at least as good as learning from scratch. Next, we consider target tasks with varying dynamics. The source environment has the standard gym values for damping and the four target environments have different values of damping increased gradually in each task. The least similar task has the highest damping values in the joints.

**Ant-v3:** This is also a high dimensional continuous control task where the goal is to make an ant-robot move in the forward direction by applying torques on the hinges that connect each leg and torso of the robot. The observation space is 27-dimensional and the action space, $a \in \mathbb{R}^8 \forall a \in \mathcal{A}$ where each action is a torque applied at the hinge joints with a value between $[-1.0, 1.0]$. The source environment has the standard gym robot and the four target environments have varied dynamics by changing the leg lengths of the robot. Representative figures of these dynamics can be found in the appendix.

For each environment, the knowledge transferred is the source optimal policy, data collected from the target task is used as the target knowledge, and average returns during each evaluation episode are used as the performance evaluation metric, $\Lambda_i : \pi_{\mathcal{S}}^* \times \mathcal{D}_{\mathcal{T},t} \to \mathbb{E}^{\pi_{\mathcal{T},t}} [\sum_k r_k]$. To obtain the source optimal policy, the SAC algorithm is utilized to train a policy from scratch in each source environment. In each of these experiments, we perform and compare our algorithm APT-RL against one of the recent benchmarks on transfer RL, the REPAINT algorithm proposed by Tao et al. (2020). We also compare APT-RL against zero-shot source policy transfer and SAC without any source knowledge. Finally, to show how our proposed transferability metrics can unify existing strategies, we calculate several performance metrics from the literature (Taylor et al., 2007). Specifically, we calculate the following: area under the reward curve $\Delta_T$ at the end of each experiment that quantifies the total reward, number of samples $n_{\text{threshold}}$ needed to reach 90% of the maximum reward value when learning from scratch and number of samples $n_\infty$ to reach steady state reward value of learning from scratch. We calculate the steady state reward value by taking an average over the last $n$-timesteps of the reward values when learning from scratch and $n_\infty$ is the number of samples required to obtain 99% of this value.

## 6 Experiment Results

### 6.1 Task similarity

We leverage Algorithm 2 to calculate the task similarity between the source and each of the target tasks. The empirical task similarity is shown in Fig. 2. For the pendulum environment in Fig. 2(a), the target tasks become gradually less similar to the source. This aligns with the intuitive sense as the gravity of the target environments increases gradually from 10.0 to 18.0. It is also easy to see that the source models become less accurate and the variance of the prediction also increases gradually. Fig. 2(b) shows the similarity in tasks for the half-cheetah environment with varying rewards. For reward similarity, we can see the highest dissimilarity when the robot is provided a negative reward instead of a positive reward. The source and the target task are least similar when the forward reward is opposite. Similarly, for the half-cheetah environment with varying dynamics in Fig. 2(c), we can see an approximately linear trend in the similarity of the dynamics. This makes sense as the change in joint damping is gradual and constant. Finally, in Fig. 2(d) we show the task similarity in the Ant environment with varying dynamics. As we change the dynamics of each of the target environments by changing the length of the legs of the robot, the similarity between each target and the source task reduces monotonically.

### 6.2 Transferability of APT-RL, $\Lambda_{\text{APT-RL}}$

In the following, we demonstrate the effectiveness of the proposed Algorithm 1, APT-RL, utilizing the proposed transferability metrics, and also investigate its relationship with task similarity. Note that we show results for three out of the four target tasks in the pendulum environment as none of the algorithms was able to solve the problem for $g = 18.0$. We also omit the results from the benchmark REPAINT algorithm for the pendulum environments, as it failed to solve the problem within the allocated number of interactions.

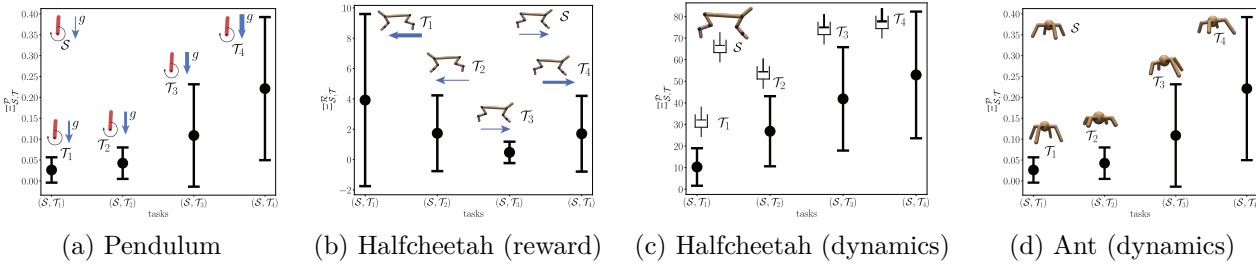

(a) Pendulum     (b) Halfcheetah (reward)     (c) Halfcheetah (dynamics)     (d) Ant (dynamics)

Figure 2: **Task similarity:** Empirical task similarity between several variations of pendulum, Half-cheetah and Ant environments

.

### 6.2.1   Pendulum-v3

For this environment, APT-RL outperforms both learning from scratch and zero-shot policy transfer by a large margin in terms of performance evaluation metric and relative transfer performance. To demonstrate the importance of synchronous source policy improvement with target data, we also show results for an APT-RL with a fixed source policy. As shown in Fig. 3, APT-RL with synchronous source policy update converges quickly in terms of average return for each of the target tasks. To obtain a similar asymptotic evaluation performance metric APT-RL uses only 20 evaluation episodes whereas learning from scratch requires 60 episodes. Thus APT-RL is almost three times faster in this case for obtaining a similar evaluation metric as learning from scratch. This is also evident from the relative transfer performance metric $\tau$ for each of the target tasks. The large jumps of $\tau$ in the positive y-direction can be interpreted as positive relative transfer performance. Intuitively, this means that the transfer strategy improves learning performance in the target task. Additionally, the learning performance seems to increase with more task similarity. This makes sense as it becomes more challenging to utilize the source knowledge in the target task if the tasks have lower similarities between them. We also show the value of the regularization temperature parameter, $\beta_t$, over time when learning in the target task using APT-RL. When the tasks have lower similarity, the temperature parameter is found to have a smaller value. Also, the temperature parameter decreases over time as the target policy gets better. This makes sense because we do not want to leverage the source knowledge when the target policy is better than the source policy. Additional evaluation metrics can be found in Table 2 where we show that Apt-RL makes more than 39% improvement in each of the target tasks in terms of area under the reward curve $\Delta_T$. Apt-RL is also more than 30% sample efficient to reach 90%-reward threshold of learning from scratch and 10% more sample efficient to reach steady state reward value of learning from scratch.

### 6.2.2   Half-cheetah-v3

Fig. 4 (a)-(c) shows the transfer evaluation performance for three target tasks with varying dynamics and Fig. 4(d) shows the performance for one target task with negative reward. We observe similar trends, compared to the pendulum tasks, in learning performance and changes in temperature parameter value when learning in the target tasks. In most cases, APT-RL can not only learn faster but also achieve higher average returns than learning from scratch as shown in Fig. 4. Most importantly, APT-RL performs as good as learning from scratch for the target task with varying rewards. Note that, this is an adversarial source task as the robot needs to learn to run in the opposite direction in the target task. In contrast, the REPAINT algorithm fails to achieve similar evaluation performance and in most cases, obtains evaluation performance lower than learning from scratch using SAC. As REPAINT is a PPO-based on-policy algorithm, this result aligns with the previously reported performance of SAC and PPO algorithms (Haarnoja et al., 2018). The impressive performance of APT-RL may be explained by the fact that the source optimal policy jumpstarts the target policy. This increase in learning performance, in turn, provides a positive relative transfer metric over time as shown in Fig. 5. Note that $\tau$ remains positive even after one million timesteps which means that APT-RL achieves higher learning performance than learning from scratch. Temperature parameter decreases quickly initially, then increases slightly and stays approximately constant over time.

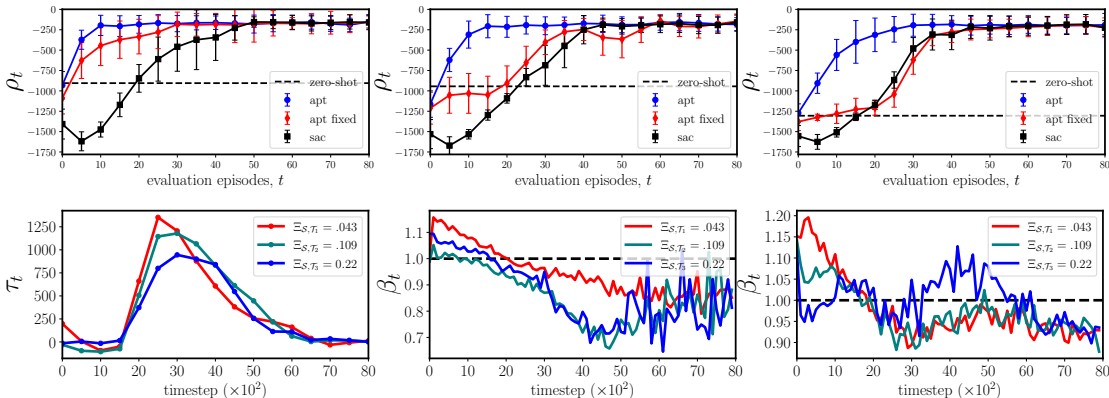

Figure 3: **APT-RL transferability, $\Lambda_{\mathbf{APT\text{-}RL}}$ in Pendulum tasks: top row**: $\rho_t$ in the several pendulum tasks are shown against vanilla sac (learning from scratch) and zero-shot policy, here average return during evaluation episode is taken as $\rho_t$, meaning $\rho_t = \mathbb{E}^{\pi_{\mathcal{T}_i}^*}[\sum_k r_k]$, results are shown within one standard deviation range, **bottom row (from left to right):** $\tau_t$ for three different tasks are shown with a corresponding mean similarity score, temperature parameter $\beta_t$ for three different tasks are shown in the next two plots; first fixed source policy and second, source policy updated with target data.

This can be explained by the fact that more weight is put into the regularization loss initially and once the target policy becomes better the effect reduces. As we keep utilizing the target data to update the source policy, the source policy improves over time and provides useful information during the later timesteps. Finally, we observe that the effect of the source policy reduces as task similarity decreases between tasks. In Table 2 we show that Apt-RL makes at least 2% improvement in each of the target tasks $\mathcal{T}_1, \mathcal{T}_2, \mathcal{T}_3$ in terms of area under the reward curve $\Delta_T$. Additionally, Apt-RL is more than 15% sample efficient to reach 90%-reward threshold of learning from scratch and 59% more sample efficient to reach steady state reward value of learning from scratch. Also, the performance of Apt-RL is consistent to learning from scratch in task $\mathcal{T}_4$ which is an adversarial task and our goal is to perform at least as good as learning from scratch.

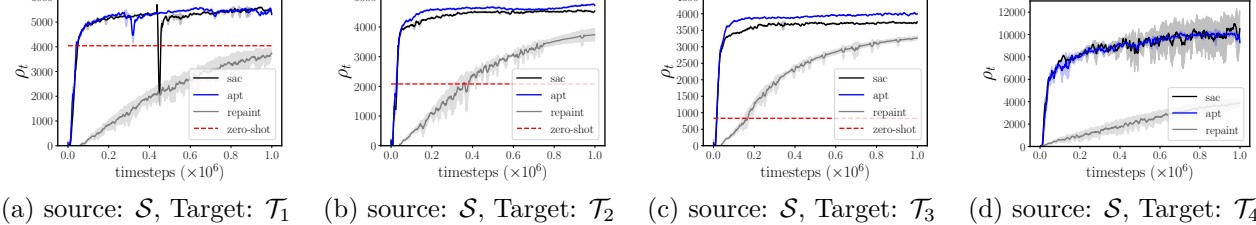

(a) source: $\mathcal{S}$, Target: $\mathcal{T}_1$    (b) source: $\mathcal{S}$, Target: $\mathcal{T}_2$    (c) source: $\mathcal{S}$, Target: $\mathcal{T}_3$    (d) source: $\mathcal{S}$, Target: $\mathcal{T}_4$

Figure 4: **APT-RL transferability, $\Lambda_{\mathbf{APT\text{-}RL}}$ in Halfcheetah tasks:** $\rho_t$ in the halfcheetah tasks with varying dynamics are shown in (a)-(c) and varying reward is shown in (d). Note that (d) is an adversarial task and the goal is to perform at least as good as learning from scratch. Apt-RL is compared against vanilla sac (learning from scratch), repaint and zero-shot policy. Average return during the evaluation episode is taken as $\rho_t$, meaning $\rho_t = \mathbb{E}^{\pi_{\mathcal{T}_i}^*}[\sum_t r_k]$. Results are shown with one standard deviation range.

### 6.2.3 Ant-v3

For the ant environment, we observe significant performance gain of APT-RL in the target task against learning from scratch, zero-shot policy transfer, and the REPAINT algorithm Fig. 6. For target tasks that are very similar to the source task, we observe a fast convergence of the policy in the target. For less similar source and target tasks, APT-RL can even achieve higher learning performance than learning from scratch. This might happen due to the jumpstart of the target policy and also due to the synchronous improvement of the source policy. The latter characteristic of APT-RL accelerates the policy update. Similar to the

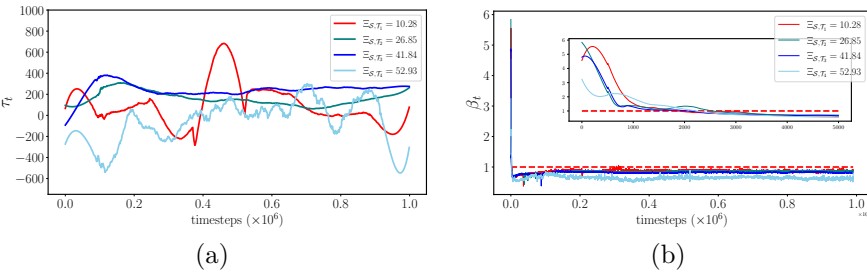

(a)                 (b)

Figure 5: (a) Relative transfer performance, $\tau_t$, for four different halfcheetah tasks are shown with corresponding mean similarity scores, (b) regularization co-efficient, $\beta_t$, is shown for four halfcheetah tasks with corresponding mean similarity scores.

half-cheetah environment, we observe that the temperature parameter decreases with less task similarity, Fig. 6.2.3. In all of these examples, APT-RL outperforms the REPAINT algorithm. Finally, Table 2 shows that Apt-RL makes more than 11% improvement in each of the target tasks $\mathcal{T}_1, \mathcal{T}_2, \mathcal{T}_3, \mathcal{T}_\triangle$ in terms of area under the reward curve $\Delta_T$. Apt-RL is more than 15% sample efficient to reach the 90%-reward threshold in all of the tasks except $\mathcal{T}_1$ where it performs almost 8% worse. Additionally, Apt-RL is more than 22% sample efficient to reach the steady state reward in all targe tasks except $\mathcal{T}_1$ where it is almost 5% more sample efficient than learning from scratch. We anticipate that the dynamics of the environment in $\mathcal{T}_1$ makes it relatively difficult to learn when compared to the rest of the target tasks.

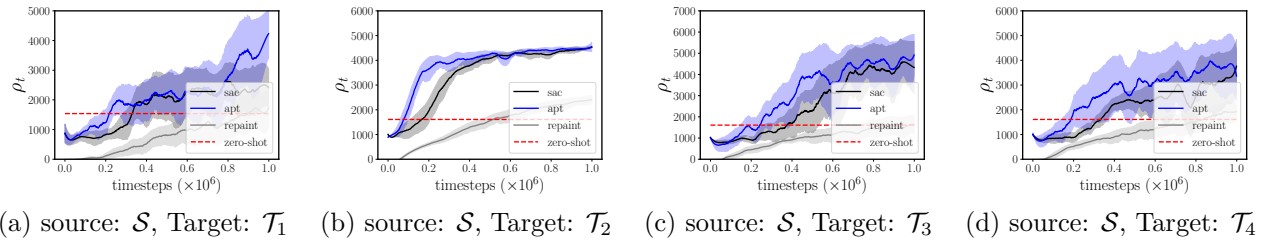

(a) source: $\mathcal{S}$, Target: $\mathcal{T}_1$    (b) source: $\mathcal{S}$, Target: $\mathcal{T}_2$    (c) source: $\mathcal{S}$, Target: $\mathcal{T}_3$    (d) source: $\mathcal{S}$, Target: $\mathcal{T}_4$

Figure 6: **APT-RL transferability, $\Lambda_{\textbf{APT-RL}}$ in Ant tasks:** $\rho_t$ in the several Ant tasks are shown in (a)-(d) where it is compared against vanilla sac (learning from scratch) and zero-shot policy. Average return during the evaluation episode is taken as $\rho_t$, meaning $\rho_t = \mathbb{E}^{\pi^*_{\mathcal{T}_i}}[\sum_k r_k]$. Results are shown within one standard deviation range.

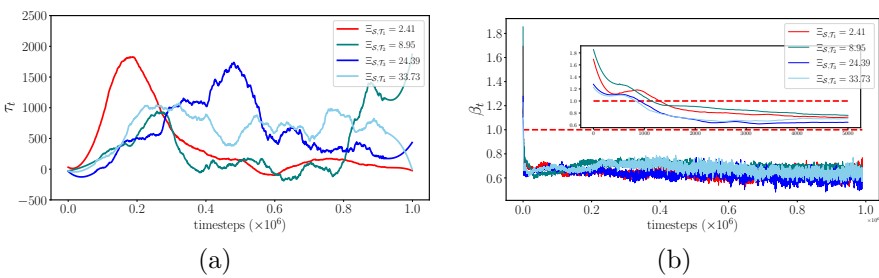

(a)                 (b)

Figure 7: (a) Relative transfer performance, $\tau_t$, for four different ant tasks are shown with corresponding mean similarity scores, (b) regularization co-efficient, $\beta_t$, is shown for four ant tasks with corresponding mean similarity scores.

| Task | | $\rho_t = \Delta_T$ | $\rho_t = n_{\text{threshold}}$ | $\rho_t = n_\infty$ |
|------|------|------|------|------|
| | $\mathcal{T}_1$ | 50.17% | 69.49% | 65.38% |
| pendulum | $\mathcal{T}_2$ | 48.23% | 30.0% | 19.67% |
| | $\mathcal{T}_3$ | 39.33% | 47.14% | 9.76% |
| | $\mathcal{T}_1$ | 1.98% | 15.86%($\downarrow$) | 76.03% |
| halfcheetah | $\mathcal{T}_2$ | 3.96% | 46.72% | 69.34% |
| | $\mathcal{T}_3$ | 6.46% | 43.48% | 59.85% |
| | $\mathcal{T}_4$ | 0.68%($\downarrow$) | 1.09% | 3.79% |
| | $\mathcal{T}_1$ | 20.90% | 8.65%($\downarrow$) | 4.67% |
| ant | $\mathcal{T}_2$ | 11.40% | 15.97% | 48.089% |
| | $\mathcal{T}_3$ | 22.56% | 24.89% | 40.14% |
| | $\mathcal{T}_4$ | 29.95% | 29.56% | 22.09% |

Table 2: Performance evaluation metric of Apt-RL that unifies previous approaches from literature: $\rho_t = \Delta_T$ is the area under the reward curve at the final timestep, $T$, $\rho_t = n_{\text{threshold}}$ is the number of samples required to reach 90% of maximum reward obtained by learning scratch and $\rho_t = n_\infty$ is the number of samples required to reach 99% of the steady state reward value of learning from scratch. Note that the downward arrow represents a percentage decrease in $\rho_t$.

# 7 Conclusion

In this paper, we proposed the APT-RL algorithm to transfer knowledge from a source task in an off-policy fashion. Through advantage-based regularization, our algorithm does not require any heuristic or manual fine-tuning of the objective function. We also introduced a new relative transfer performance metric, which can help evaluate and compare transfer learning approaches in RL. We also provided a simple, theoretically-backed algorithm to calculate task similarity, and demonstrated the alignment of our proposed transfer performance metric with source and target task similarities. We demonstrated the effectiveness of APT-RL in continuous control tasks and showed its superior performance against benchmark transfer RL algorithms. Future directions may include considering similar concepts for multi-task transfer learning scenarios, as well as benchmarking the performance of various transfer learning algorithms with the help of the transferability metrics introduced in this paper.

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
