# OpenReview forum: "An advantage based policy transfer algorithm for reinforcement learning with metrics of transferability"
_TMLR — Rejected by TMLR_

### Review · Reviewer_F9xj · 2023-12-19

**Summary Of Contributions:**

Contributions:
1. This paper presents a (fixed domain) transfer RL algorithm that adds a policy distillation term to the standard SAC algorithm. The policy distillation term’s coefficient is set dynamically (“advantage based”), eliminating the need for an extra hyperparameter.

2. The authors propose formal and general metrics to measure transferability and task similarity.

3. Experiments show the proposed algorithm is effective and the proposed metrics are meaningful.

**Audience:**

Yes

**Claims And Evidence:**

No

**Requested Changes:**

Required changes
* Please clarify/justify the main algorithm. (see my questions on $J_2$ and $\beta_t$)
* Please add fine-tuning the source policy as a baseline (or justify why this is not needed).
* Please address my concern of using REPAINT as a baseline, as the current setup is misleading.

Nice to have:
* Ablate with constant $\beta$.
* More challenging continuous control environments (e.g. humanoid).
* Discuss limitations of using random policy to compute task similarity.
* Add confidence intervals to some subplots (e.g. $\tau_t$).

Minor:
* Please standardize your plots. (e.g. fig 3 has a different style compared to fig 5 and fig 7)
* I see little value of table 2. Not sure what it tries to convey. Please consider make the points clearer or remove it.

**Strengths And Weaknesses:**

Strength:
* The proposed algorithm (APT-RL) is intuitive and well-motivated by the toy example. The regularization term (distillation) requires no hyperparameter tuning. These merits make the algorithm potentially attractive to practitioners.

Weaknesses:
* The main algorithm (APT-RL) needs clarifications (or justifications/corrections):
  * Eq 3, definition of $J_2$: Does this require an expectation over state $s$? (just like $J_1$)
  * Eq 4, definition of $\beta_t$: in theory, the Q function should be the optimal Q-value $Q^*$ for the target environment (similarly for the state value V function). Of course,  $Q^*$ is not available, so an approximation $Q_{\theta}$ is used. This Q-value is associated with the learned policy $\mu_{\psi}$. In general, it makes little sense to use $Q^{\mu_{\psi}}$ with actions from $\pi_{\phi}$. Due to these nuances, current definition of $\beta_t$ is deviated from the intuition stated earlier in the paper ("... have more weight when an average action taken according to $\pi_{\phi}$ is better than a random action").
* Task similarity measurement: the authors propose to use a *random policy* to collect data and fits a reward model and a dynamics model. However, for complex environments, random policy is rarely sufficient for model fitting. This makes me wonder how applicable this metric is beyond simple environments.
* Experiments:
  * REPAINT is not a proper baseline (as-is). As the author pointed out, REPAINT is a PPO based on-policy algorithm. SAC is known to be more sample efficient than PPO on these tasks. Therefore, the comparison between REPAINT and APT-RL does not mean APT-RL is more effective *as a transfer RL algorithm*. Maybe a fairer comparison is to compare REPAINT and PPO from scratch, and derive the transferability metric (as introduced in this paper). Would the author observe larger gain in APT+SAC vs SAC compared to REPAINT+PPO vs PPO?
  * Other baselines to consider: The plots contain zero-shot performance. How about fine-tuning performance?
  * It would be great to see results on other Mujoco continuous control environments (e.g. humanoid)
  * $\beta_t$ is close to 1 for all three environments (especially when synchronous source policy update is enabled). This suggests: 1) the anchoring effect of the base policy is very strong; 2) Maybe $\beta$ can just be set as a hyperparameter and the algorithm is not sensitive on it. While this would introduce a hyperparameter, it offers a tradeoff for simpler implementation.

---

### Review · Reviewer_Hsjo · 2024-02-11

**Summary Of Contributions:**

The paper proposes a novel off-policy transfer learning algorithm called Advantage-based Policy Transfer Algorithm (ATP-RL). This algorithm uses the advantage function to relatively weight the influence of the source policy (optimal/near-optimal policy obtained from training on the source task) on the target task. The advantages of ATP-RL is demonstrated on different continuous control experiments (Pendulum and MuJoCo), as compared to a set of recent baselines in transfer RL. The paper considers fixed domain environments where the source and target task share the same state space and action space.

**Audience:**

Yes

**Claims And Evidence:**

No

**Requested Changes:**

Please address my comments in the Weaknesses above.

**Strengths And Weaknesses:**

Strengths:

1. The paper is (mostly) well-written and well-organized.

2. The toy problem introduced in Figure 1 is a nice running example. This example helps considerably with the understanding of the paper.

3. The paper is well-situated in literature and does a thorough job of referencing the important related works in the area.

4. The experiments are comprehensive and nicely showcases the advantages of the ATP-RL as compared to some strong baselines.

Weaknesses

1. I think that the paper's claim of introducing a new metric is incomplete. A metric in measure theoretic terms needs to satisfy several properties in the associated metric space (for example: distance to self is 0, positivity, symmetry and triangle inequality), however the paper does not prove any of these properties for the transferability metric.

2. Given my first point, I think the Theorem 1 depends on the metric and hence the existence of this metric needs to be proved before Theorem 1 can hold.

3. I am unable to appreciate the significance of Theorem 2. Theorem 2 says that the bound on the differences in Q values between the source and target tasks is a function of their reward difference? Is this not obvious, given the fact that the Q function captures the expected discounted cumulative rewards in the first place?

4. I am not sure if the synchronous updates of the source policy will lead to an improvement in sample efficiency in all cases. Though the paper has shown some examples in which it does improve the sample efficiency, sometimes this method may lead to requiring more samples than doing just updates to the target policy. If the target task has important differences from the source task, the new data coming from synchronous updates may just be adding noise on the source policy which slows the learning of a target policy using the source policy. Are there any theoretical results that can demonstrate an improvement in sample efficiency while using the synchronous updates?

5. The paper mentions that it does not require the learning of the optimal source policy in the Related Works section. However, in Section 3.1, the paper mentions that the current policy is biased to stay close to the optimal source policy by minimizing the cross-entropy. If the optimal source policy is not available, how is it possible to calculate this cross-entropy?

6. I did not understand the legends in Figure 5 and 7. In the legend it is mentioned that these are the mean of the similarity scores. What does having a lower mean similarity score indicate and what characteristic of APT-RL is captured in these graphs? There is no explanation in the text for Figure 7, and more explanation is required for Figure 5.

7. Some important baselines are missing. The paper should compare to Zhang et al. 2018 and Gupta et al. 2017, or include strong reasons for why such comparisons are unnecessary.

Minors: The references to the appendices are broken throughout the paper and need to be fixed.

---

### Review · Reviewer_chQa · 2024-02-19

**Summary Of Contributions:**

The paper considers the Transfer Reinforcement Learning setting in which knowledge from a source task has to be transferred to a target task. The main contribution of the paper is to consider that the target task is learned by a Soft Actor-Critic agent, and that this agent's loss can be extended to encourage the learned policy to remain close to the optimal policy learned on the source task. The weight of that additional loss term is computed every gradient step, in a state-dependent way.

Theoretical results are given in the paper and the empirical evaluation on challenging environments demonstrates that the idea is encouraging and allows for better initial performance, higher sample-efficiency and equal final performance compared to SAC without transfer, and some other transfer technique.

**Audience:**

Yes

**Broader Impact Concerns:**

There does not seem to be concerns about the broader impact of this work.

**Claims And Evidence:**

Yes

**Requested Changes:**

The paragraph above Equation 4, the one that introduces $\beta_t$, does not make much sense and is difficult to understand. Why would $\beta$ need to be higher (thus steering the actor more forwards the source task policy) when an action executed by the actor is better than random (and not the opposite?). It seems that there is either unclear wording, or a mix-up between the source and target policies, or some confusion about whether $\beta$ should be higher or lower.

**Strengths And Weaknesses:**

Strengths:

- While the abstract and introduction of the paper feel flimsy and should be reworked a bit (to be more concise and more scientific), the rest of the paper is clearly written, presents the contribution well, and is overall pleasant to read
- The contribution is simple and well-motivated. It also appears to be quite easy to implement.
- The results are very encouraging

Weaknesses:

- The empirical evaluation, albeit thorough regarding what "happens" to the algorithms, does not consider that many related transfer learning techniques. For instance, it does not seem that the simple "initialize the weights of the target agent to the weights of the source agent, then continue training from that" approach is evaluated. Literature on Policy Shielding could also be relevant to evaluate. For instance, some methods train the target agent from scratch, but with one extra action: execute the optimal action from the source task. This also allows the target agent to get help from the source agent, while still allowing it to outperform the source task's policy.

---

### Review · Reviewer_25bT · 2024-02-21

**Summary Of Contributions:**

The paper presents a methods for online transfer learning APT-RL, which is based on two mechanisms. Transferring from a source policy and synchronous parameters updates. Moreover, the paper proposed a transferability metrics and shows its theoretical and empirical properties.

**Audience:**

Yes

**Claims And Evidence:**

No

**Requested Changes:**

See above

**Strengths And Weaknesses:**

I find the topic of the paper very interesting, however, I have major concerns regarding the claims presented. I'd be happy to review them upon the authors responses.

I am not sure what the core principle of Algorithm 1 is. The policy update goes along (1) with $\beta_t$ being adapted. The standard SAC loss (2) updates the policy to increase the $Q$-function (advantage). The additional new objective (3) improves the policy, again, based on the advantage (i.e. the source policy is cloned only if its advantage is high). It feels like these two mechanism overlap. Perhaps the source policy is cloned only if it is similar to the advantage?)

A more nuanced version of this concern, and perhaps a good experimental idea, would be to test various choices of the weighting function (instead only exp in (4)).

I find it troubling not to see a fine-tuning baseline, i.e. just continue training on the target task, perhaps with using an additional fresh head ([1] makes a detailed study of the transfer components).

I find the empirical results mildly convincing. The proposed method is not that much better as training from scratch. Additionally, it would be instructive to the comparison of APT-RL with the one for which $\beta$ is fixed.

Please note how many seeds are used in the empirical evalutions.

---

### Decision · Action_Editor_HgiM · 2024-04-16

**Recommendation:** Reject

**Comment:**

It very well could be the case that the reviewers (and myself) are mistaken about the lack of convincing evidence for the paper's claims (though they are shared and numerous enough that I think a complete reversal extremely unlikely). But the authors' lack of engagement in the review process makes the rejection decision straightforward.

**Audience:**

If the issues wrt claims and evidence were resolved, then the scope and content of the paper would definitely be of broad interest to TMLR's audience.

**Claims And Evidence:**

Unfortunately, it is unclear whether the theoretic claims holds as some reviewers dispute the relevant conditions. All of the reviewers were also unconvinced by the empirical evidence, with the lack of a simple fine-tuning baseline casting doubt on the relative efficacy of the author's approach.

**Resubmission Of Major Revision:**

The authors may consider submitting a major revision at a later time.